# Mapping distinct timescales of functional interactions among brain networks

**Mali Sundaresan**[1]
s.malisundar@gmail.com

**Arshed Nabeel**[2]
arshed@iisc.ac.in

**Devarajan Sridharan**[1,2]*
sridhar@iisc.ac.in

[1]Center for Neuroscience, Indian Institute of Science, Bangalore
[2]Department of Computer Science and Automation, Indian Institute of Science, Bangalore

## Abstract

Brain processes occur at various timescales, ranging from milliseconds (neurons) to minutes and hours (behavior). Characterizing functional coupling among brain regions at these diverse timescales is key to understanding how the brain produces behavior. Here, we apply instantaneous and lag-based measures of conditional linear dependence, based on Granger-Geweke causality (GC), to infer network connections at distinct timescales from functional magnetic resonance imaging (fMRI) data. Due to the slow sampling rate of fMRI, it is widely held that GC produces spurious and unreliable estimates of functional connectivity when applied to fMRI data. We challenge this claim with simulations and a novel machine learning approach. First, we show, with simulated fMRI data, that instantaneous and lag-based GC identify distinct timescales and complementary patterns of functional connectivity. Next, we analyze fMRI scans from 500 subjects and show that a linear classifier trained on either instantaneous or lag-based GC connectivity reliably distinguishes task versus rest brain states, with ∼80-85% cross-validation accuracy. Importantly, instantaneous and lag-based GC exploit markedly different spatial and temporal patterns of connectivity to achieve robust classification. Our approach enables identifying functionally connected networks that operate at distinct timescales in the brain.

## 1 Introduction

Processes in the brain occur at various timescales. These range from the timescales of milliseconds for extremely rapid processes (e.g. neuron spikes), to timescales of tens to hundreds of milliseconds for processes coordinated across local populations of neurons (e.g. synchronized neural oscillations), to timescales of seconds for processes that are coordinated across diverse brain networks (e.g. language) and even up to minutes, hours or days for processes that involve large-scale neuroplastic changes (e.g. learning a new skill). Coordinated activity among brain regions that mediate each of these cognitive processes would manifest in the form of functional connections among these regions at the corresponding timescales. Characterizing patterns of functional connectivity that occur at these different timescales is, hence, essential for understanding how the brain produces behavior.

Measures of linear dependence and feedback, based on Granger-Geweke causality (GC) [10][11]), have been used to estimate instantaneous and lagged functional connectivity in recordings of brain activity made with electroencephalography (EEG, [6]), and electrocorticography (ECoG, [3]). However, the application of GC measures to brain recordings made with functional magnetic resonance imaging (fMRI) remains controversial [22][20][2]. Because the hemodynamic response is produced and sampled at a timescale (seconds) several orders of magnitude slower than the underlying neural processes (milliseconds), previous studies have argued that GC measures, particularly lag-based GC, produce spurious and unreliable estimates of functional connectivity from fMRI data [22][20].

---

Three primary confounds have been reported with applying lag-based GC to fMRI data. First, systematic hemodynamic lags: a slower hemodynamic response in one region, as compared to another could produce a spurious directed GC connection from the second to the first [22] [4]. Second, in simulations, measurement noise added to the signal during fMRI acquisition was shown to produce significant degradation in GC functional connectivity estimates [20]. Finally, downsampling recordings to the typical fMRI sampling rate (seconds), three orders of magnitude slower than the timescale of neural spiking (milliseconds), was shown to effectively eliminate all traces of functional connectivity inferred by GC [20]. Hence, a previous, widely cited study argued that same-time correlation based measures of functional connectivity, such as partial correlations, fare much better than GC for estimating functional connectivity from fMRI data [22].

The controversy over the application of GC measures to fMRI data remains unresolved to date, primarily because of the lack of access to "ground truth". On the one hand, claims regarding the efficacy of GC estimates based on simulations, are only as valid as the underlying model of hemodynamic responses. Because the precise mechanism by which neural responses generate hemodynamic responses is an active area of research [7], strong conclusions cannot be drawn based on simulated fMRI data alone. On the other hand, establishing "ground truth" validity for connections estimated by GC on fMRI data require concurrent, brain-wide invasive neurophysiological recordings during fMRI scans, a prohibitive enterprise.

Here, we seek to resolve this controversy by introducing a novel application of machine learning that works around these criticisms. We estimate instantaneous and lag-based GC connectivity, first, with simulated fMRI time series under different model network configurations and, next, from real fMRI time series (from 500 human subjects) recorded under different task conditions. Based on the GC connectivity matrices, we train a linear classifier to discriminate model network configurations or subject task conditions, and assess classifier accuracy with cross validation. Our results show that instantaneous and lag-based GC connectivity estimated from empirical fMRI data can distinguish task conditions with over 80% cross-validation accuracies. To permit such accurate classification, GC estimates of functional connectivity must be robustly consistent within each model configuration (or task condition) and reliably different across configurations (or task conditions). In addition, drawing inspiration from simulations, we show that GC estimated on real fMRI data downsampled to 3x-7x the original sampling rate provides novel insights into functional brain networks that operate at distinct timescales.

## 2 Simulations and Theory

### 2.1 Instantaneous and lag-based measures of conditional linear dependence

The linear relationship among two multivariate signals $\mathbf{x}$ and $\mathbf{y}$ conditioned on a third multivariate signal $\mathbf{z}$ can be measured as the sum of linear feedback from $\mathbf{x}$ to $\mathbf{y}$ ($F_{x \to y}$), linear feedback from $\mathbf{y}$ to $\mathbf{x}$ ($F_{y \to x}$), and instantaneous linear feedback ($F_{x \circ y}$) [11][16]. To quantify these linear relationships, we model the future of each time series in terms of their past values with a well-established multivariate autoregressive (MVAR) model (detailed in Supplementary Material, Section S1).

Briefly, $F_{x \to y}$ is a measure of the improvement in the ability to predict the future values of $\mathbf{y}$ given the past values of $\mathbf{x}$, over and above what can be predicted from the past values of $\mathbf{z}$ and $\mathbf{y}$, itself (and vice versa for $F_{y \to x}$). $F_{x \circ y}$, on the other hand, measures the instantaneous influence between $\mathbf{x}$ and $\mathbf{y}$ conditioned on $\mathbf{z}$ (see Supplementary Material, Section S1). We refer to $F_{x \circ y}$, as instantaneous GC (iGC), and $F_{x \to y}$ $F_{y \to x}$ as lag-based GC or directed GC (dGC), with the direction of the influence ($\mathbf{x}$ to $\mathbf{y}$ or vice versa) being indicated by the arrow. The "full" measure of linear dependence and feedback $F_{x,y}$ is given by :

$$F_{x,y} = F_{x \to y} + F_{y \to x} + F_{x \circ y} \tag{1}$$

$F_{x,y}$ measures the complete conditional linear dependence between two time series. If, at a given instant, no aspect of one time series can be explained by a linear model containing all the values (past and present) of the other, $F_{x,y}$ will evaluate to zero [16]. These measures are firmly grounded in information theory and statistical inferential frameworks [9].

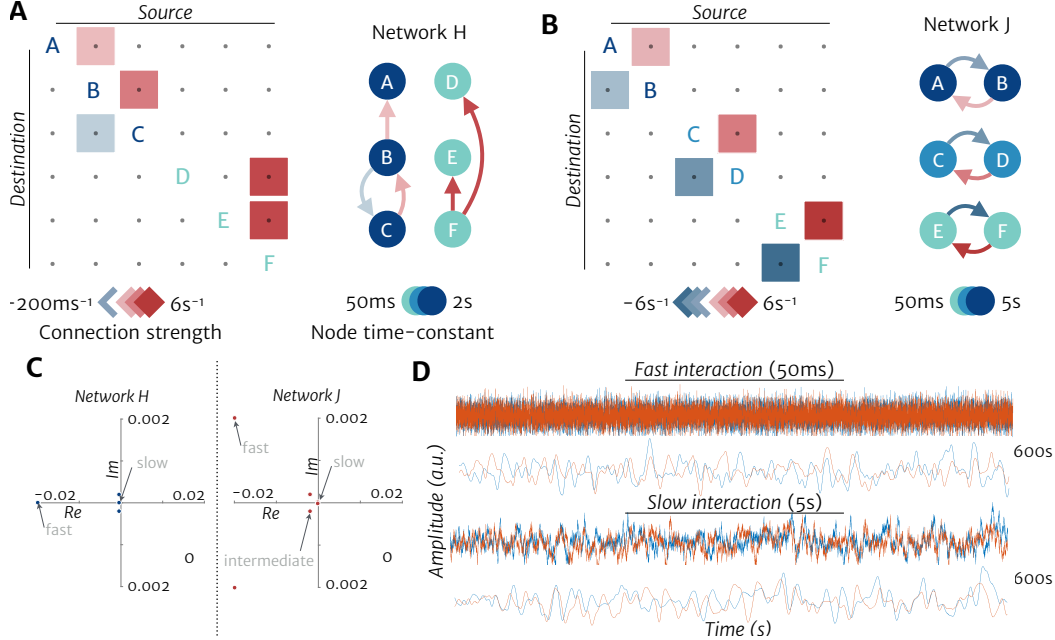

Figure 1: **Network simulations**. (A) Network configuration *H*. (Left) Connectivity matrix. Red vs. blue: Excitatory vs. inhibitory connections. Deeper hues: Higher connection strengths. Non-zero value at $(i, j)$ corresponds to a connection from node $j$ to node $i$ (column to row). Sub-network A-B-C operates at a fast timescale (50 ms) whereas D-E-F operates at a slow timescale (2 s). (Right) Network schematic showing the connectivity matrix as a graph. (B) Network configuration *J*. Conventions are the same as in A. (C) The eigenspectra of networks *H* (left) and *J* (right). (D) Simulated time series in network configuration *J* with fast (top panel) and slow (bottom panel) dynamics, corresponding to nodes A-B and E-F, respectively. Within each panel, the top plot is the simulated neural time series, and the bottom plot is the simulated fMRI time series.

## 2.2 Simulating functional interactions at different timescales

To test the ability of GC measures to reliably recover functional interactions at different timescales, we simulated fMRI time series for model networks with two configurations of directed connectivity. Simulated fMRI time series were generated using a two-stage model (2): the first stage involved a latent variable model that described neural dynamics, and the second stage that convolved these dynamics with the hemodynamic response function (HRF) to obtain the simulated fMRI time series.

$$\dot{\mathbf{x}} = A\mathbf{x} + \varepsilon \qquad\qquad\qquad \mathbf{y} = H * \mathbf{x} \qquad\qquad (2)$$

where $A$ is the neural ("ground truth") connectivity matrix, $\mathbf{x}$ is the neural time series, $\dot{\mathbf{x}}$ is $d\mathbf{x}/dt$, $H$ is the canonical hemodynamic response function (HRF; simulated with *spm_hrf* in SPM8 software), $*$ is the convolution operation, $\mathbf{y}$ is the simulated BOLD time series, and $\varepsilon$ is i.i.d Gaussian noise. Other than noise $\varepsilon$, other kinds of external input were not included in these simulations. Similar models have been employed widely for simulating fMRI time series data previously [22][2][20].

First, we sought to demonstrate the complementary nature of connections estimated by iGC and dGC. For this, we used network configuration H, shown in Fig. 1A. Note that this corresponds to two non-interacting sub-networks, each operating at distinctly different timescales (50 ms and 2000 ms node decay times, respectively) as revealed by the eigenspectrum of the connectivity matrix (Fig. 1C). For convenience, we term these two timescales as "fast" and "slow". Moreover, each sub-network operated with a distinct pattern of connectivity, either purely feedforward, or with feedback (E-I). Dynamics were simulated with a 1 ms integration step (Euler scheme), convolved with the HRF and then downsampled to 0.5 Hz resolution (interval of 2 s) to match the sampling rate (repeat time, TR) of typical fMRI recordings.

Second, we sought to demonstrate the ability of dGC to recover functional interactions at distinct timescales. For this, we simulated a different network configuration J, whose connectivity matrix

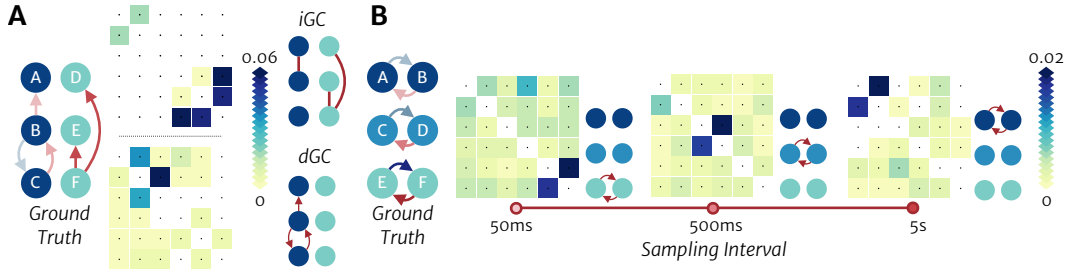

Figure 2: **Connectivity estimated from simulated data.** (A) iGC and dGC values estimated from simulated fMRI time series, network *H*. (Leftmost) Ground truth connectivity used in simulations. (Top) Estimated iGC connectivity matrix (left) and significant connections (right, p<0.05) estimated by a bootstrap procedure using 1000 phase scrambled surrogates[18]. (Bottom) Same as top panel, but for dGC. (B) dGC estimates from simulated fMRI time series, network *J*, sampled at three different sampling intervals: 50 ms (left), 500 ms (middle) and 5 s (right). In each case the estimated dGC matrix and significant connections are shown, with the same conventions as in panel (A).

is shown in Fig. 1B. This network comprised three non-interacting sub-networks operating at three distinct timescales (50 ms, 0.5 s, and 5 s node decay times; eigenspectrum in Fig. 1C). As before, simulated dynamics were downsampled at various rates – 20 Hz, 2 Hz, 0.2 Hz – corresponding to sampling intervals of 50 ms, 0.5 s, and 5 s, respectively. The middle interval (0.5 s) is closest to the repeat time (TR=0.7 s) of the experimental fMRI data used in our analyses; the first and last intervals were chosen to be one order of magnitude faster and slower, respectively.

Sufficiently long (3000 s) simulated fMRI timeseries were generated for each network configuration (H and J). Sample time series from a subset of these simulations before and after hemodynamic convolution and downsampling are shown in Fig. 1D.

## 2.3 Instantaneous and lag-based GC identify complementary connectivity patterns

Our goal was to test if the ground truth neural connectivity matrix ($A$ in equation 2) could be estimated by applying iGC and dGC to the fMRI time series $\mathbf{y}$. dGC was estimated from the time series with the MVGC toolbox (GCCA mode) [1][19] and iGC was estimated from the MVAR residuals [16].

For simulations with network configuration H, iGC and dGC identified connectivity patterns that differed in two key respects (Fig. 2A). First, iGC identified feedforward interactions at both fast and slow timescales whereas dGC was able to estimate only the slow interactions, which occurred at a timescale comparable to the sampling rate of the measurement. Second, dGC was able to identify the presence of the E-I feedback connection at the slow timescale, whereas iGC entirely failed to estimate this connection. In the Supplementary Material (Section S2), we show theoretically why iGC can identify mutually excitatory or mutually inhibitory feedback connections, but fails to identify the presence of reciprocal excitatory-inhibitory (E-I) feedback connections, particularly when the connection strengths are balanced.

For simulations with network configuration J, dGC identified distinct connections depending on the sampling rate. At the highest sampling rate (20 Hz), connections at the fastest timescales (50 ms) were estimated most effectively, whereas at the slowest sampling rates (0.2 Hz), only the slowest timescale connections (5 s) were estimated; intermediate sampling rates (2 Hz) estimated connections at intermediate timescales (0.5 s). Thus, dGC estimated robustly those connections whose process timescale was closest to the sampling rate of the data.

The first finding — that connections at fast timescales (50 ms) could not be estimated from data sampled at much lower rates (0.2 Hz) — is expected, and in line with previous findings. However, the converse finding — that the slowest timescale connections (5 s) could not be detected at the fastest sampling rates (20 Hz) — was indeed surprising. To better understand these puzzling findings, we performed simulations over a wide range of sampling rates for each of these connection timescales; the results are shown in Supplementary Figure S1. dGC values (both with and without convolution with the hemodynamic response function) systematically increased from baseline, peaked at a sampling rate corresponding to the process timescale and decreased rapidly at higher sampling rates, matching

recent analytical findings[2]. Thus, dGC for connections at a particular timescale was highest when the data were sampled at a rate that closely matched that timescale.

Two key conclusions emerged from these simulations. First, functional connections estimated by dGC can be distinct from and complementary to connections identified by iGC, both spatially and temporally. Second, connections that operate at distinct timescales can be detected by estimating dGC on data sampled at distinct rates that match the timescales of the underlying processes.

# 3    Experimental Validation

We demonstrated the success of instantaneous and lag-based GC to accurately estimate functional connectivity with simulated fMRI data. Nevertheless, application of GC measures to real fMRI data is fraught with significant caveats, associated with hemodynamic confounds and measurement noise, as described above. We asked whether, despite these confounds, iGC and dGC would be able to produce reliable estimates of connectivity in real fMRI data. Moreover, as with simulated data, would iGC and dGC reveal complementary patterns of connectivity that varied reliably with different task conditions?

## 3.1    Machine learning, cross-validation and recursive feature elimination

We analyzed minimally preprocessed brain scans of 500 subjects, drawn from the Human Connectome Project (HCP) database [12]. We analyzed data from resting state and seven other task conditions (total of 4000 scans; Supplementary Table S1). In the main text we present results for classifying the resting state from the language task; the other classifications are reported in the Supplementary Material. The language task involves subjects listening to short segments of stories and evaluating semantic content in the stories. This task is expected to robustly engage a network of language processing regions in the brain. The resting state scans served as a "task-free" baseline, for comparison.

Brain volumes were parcellated with a 14-network atlas [21] (see Supplementary Material Section S3; Supplementary Table S2). Network time series were computed by averaging time series across all voxels in a given network using Matlab and SPM8. These multivariate network time series were then fit with an MVAR model (Supplementary Material Section S1). Model order was determined with the Akaike Information Criterion for each subject, was typically 1, and did not change with further downsampling of the data (see next section). The MVAR model fit was then used to estimate both an instantaneous connectivity matrix using iGC ($F_{x \circ y}$) and a lag-based connectivity matrix using dGC ($F_{x \to y}$).

The connection strengths in these matrices were used as feature vectors in a linear classifier based on support vector machines (SVMs) for high dimensional predictor data. We used Matlab's *fitclinear* function, optimizing hyperparameters using a 5-fold approach: by estimating hyperparameters with five sets of 100 subjects in turn, and measuring classification accuracies with the remaining 400 subjects; the only exception was for the classification analysis with averaging GC matrices (Fig. 3B) for which classification was run with default hyperparameters (regularization strength = 1/(cardinality of training-set), ridge penalty). The number of features for iGC-based classification was 91 (upper triangular portion of the symmetric 14×14 iGC matrix) and for dGC-based classification was 182 (all entries of the 14×14 dGC matrix, barring self-connections on the main diagonal). Based on these functional connectivity features, we asked if we could reliably predict the task condition (e.g. language versus resting). Classification performance was tested with leave-one-out and k-fold cross-validation. We also assessed the significance of the classification accuracy with permutation testing [14] (Supplementary Material, Section S4).

Finally, we wished to identify a key set of connections that permitted accurately classifying task from resting states. To accomplish this, we applied a two-stage recursive feature elimination (RFE) algorithm [5], which identified a minimal set of features that provided maximal cross validation accuracy (generalization performance). Details are provided in the Supplementary Material (Section S5, Supplementary Figs. S2-S3).

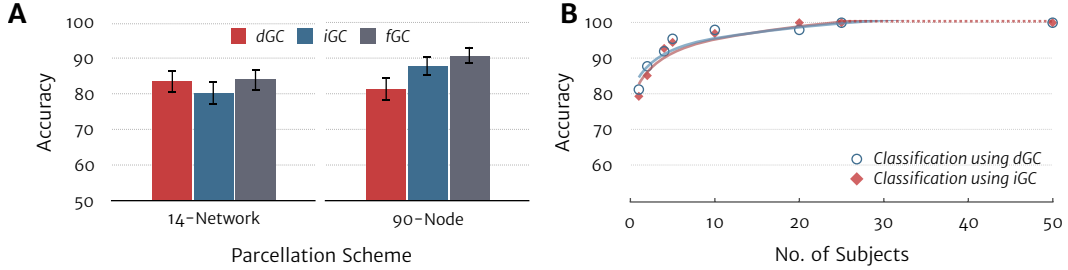

Figure 3: **Classification based on GC connectivity estimates in real data.** (A) Leave-one-out classification accuracies for different GC measures for the 14-network parcellation (left) and the 90-node parcellation (right). Within each group, the first two bars represent the classification accuracy with dGC and iGC respectively. The third bar is the classification accurcay with fGC (see equation 1). Chance: 50% (two-way classification). Error-bars: Clopper-Pearson binomial confidence intervals. (B) Classification accuracy when the classifier is tested on average GC matrices, as a function of number of subjects being averaged (see text for details).

## 3.2 Instantaneous and lag-based GC reliably distinguish task from rest

Both iGC and dGC connectivity were able to distinguish task from resting state significantly above chance (Fig. 3A). Average leave-one-out cross validation accuracy was 80.0% with iGC and 83.4% with dGC (Fig. 3A, left). Both iGC and dGC classification exhibited high precision and recall at identifying language task (precision= 0.81, recall= 0.78 for iGC and precision= 0.85, recall= 0.81 for dGC). k-fold (k=10) cross-validation accuracy was also similar for both the GC measures (79.4% for iGC and 83.7% for dGC).

dGC and iGC are complementary measures of linear dependence, by their definition. We asked if combining them would produce better classification performance. We combined dGC and iGC in two ways. First, we performed classification after pooling features (connectivity matrices) across both dGC and iGC ("iGC ∪ dGC"). Second, we estimated the full GC measure ($F_{x,y}$), which is a direct sum of dGC and iGC estimates (see equation 1). Both of these approaches yielded marginally higher classification accuracies – 88.2% for iGC ∪ dGC and 84.6% for fGC – than dGC or iGC alone.

Next, we asked if classification would be more accurate if we averaged the GC measures across a few subjects, to remove uncorrelated noise (e.g. measurement noise) in connectivity estimates. For this, the data were partitioned into two groups of 250 subjects: a training (T) group and a test (S) group. The classifier was trained on group T and the classifier prediction was tested by averaging GC matrices across several folds of S, each fold containing a few (m=2,4,5,10 or 25) subjects. Prediction accuracy for both dGC and iGC reached ∼90% with averaging as few as two subjects' GC matrices, and reached ∼100%, with averaging 10 subjects' matrices (Fig. 3B).

We also tested if these classification accuracies were brain atlas or cognitive task specific. First, we tested an alternative atlas with 90 functional nodes based on a finer regional parcellation of the 14 functional networks [21]. Classification accuracies for iGC and fGC improved (87.9% and 90.8%, respectively), and for dGC remained comparable (81.4%), to the 14 network case (Fig. 3A, right). Second, we performed the same GC-based classification analysis for six other tasks drawn from the HCP database (Supplementary Table S1) . We discovered that all of the remaining six tasks could be classified from the resting state with accuracy comparable to the language versus resting classification (Supplementary Fig. S4).

Finally, we asked how iGC and dGC classification accuracies would compare to those of other functional connectivity estimators. For example, partial correlations (PC) have been proposed as a robust measure of functional connectivity in previous studies [22]. Classification accuracies for PC varied between 81-96% across tasks (Supplementary Fig. S5B). PC's better performance is expected: estimators based on same-time covariance are less susceptible to noise than those based on lagged covariance, a result we derive analytically in the Supplementary Material (Section S6). Also, when classifying language task versus rest, PC and iGC relied on largely overlapping connections (∼60% overlap) whereas PC and dGC relied on largely non-overlapping connections (∼25% overlap; Supplementary Fig. S5C). These results highlight the complementary nature of PC and dGC connectivity. Moreover, we demonstrate, both with simulations and with real-data, that

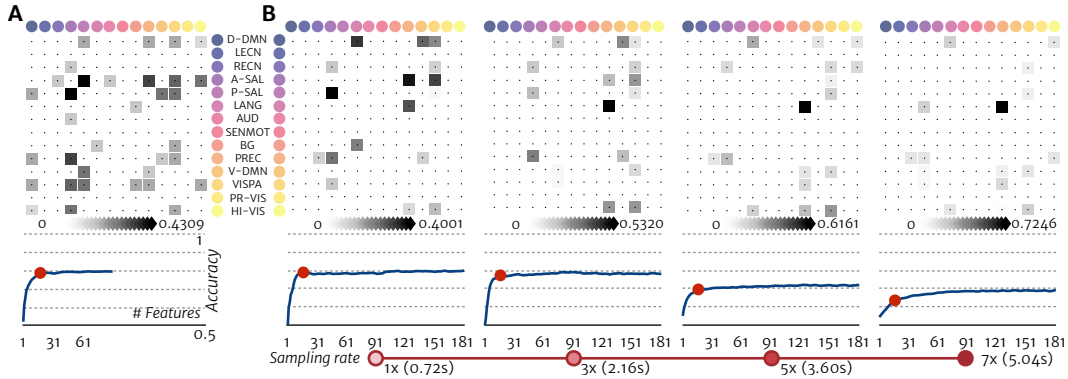

Figure 4: **Maximally discriminative connections identified with RFE** (A) (Top) iGC connections that were maximally discriminative between the language task and resting state, identified using recursive feature elimination (RFE). Darker gray shades denote more discriminative connections (higher beta weights) (Bottom) RFE curves, with classification accuracy plotted as a function of the number of remaining features. The dots mark the elbow-points of the RFE curves, corresponding to the optimal number of discriminative connections. (B) Same as in (A), except that RFE was performed on dGC connectivity matrices with data sampled at 1x, 3x, 5x, and 7x of the original sampling interval (TR=0.72 s). Non-zero value at $(i, j)$ corresponds to a connection from node $j$ to node $i$ (column to row).

classification accuracy with GC typically increased with more scan timepoints, consistent with GC being an information theoretic measure (Supplementary Fig. S6).

These superior classification accuracies show that, despite conventional caveats for estimating GC with fMRI data, both iGC and dGC yield functional connectivity estimates that are reliable across subjects. Moreover, dGC's lag-based functional connectivity provides a robust feature space for classifying brain states into task or rest. In addition, we found that dGC connectivity can be used to predict task versus rest brain states with near-perfect (>95-97%) accuracy, by averaging connectivity estimates across as few as 10 subjects, further confirming the robustness of these estimates.

### 3.3 Characterizing brain functional networks at distinct timescales

Recent studies have shown that brain regions, across a range of species, operate at diverse timescales. For example, a recent calcium imaging study demonstrated the occurrence of fast (∼100 ms) and slow (∼1 s) functional interactions in mouse cortex [17]. In non-human primates, cortical brain regions operate at a hierarchy of intrinsic timescales, with the sensory cortex operating at faster timescales compared to prefrontal cortex [13]. In the resting human brain, cortical regions organize into a hierarchy of functionally-coupled networks characterized by distinct timescales [24]. It is likely that these characteristic timescales of brain networks are also modulated by task demands. We asked if the framework presented in our study could characterize brain networks operating at distinct timescales across different tasks (and rest) from fMRI data.

We had already observed, in simulations, that instantaneous and lag-based GC measures identified functional connections that operate at different timescales (Fig. 2A). We asked if these measures could identify connections at fast versus slow timescales (compared to TR=0.72s) that were specific to task verus rest, from fMRI recordings. To identify these task-specific connections, we performed recursive feature elimination (described in Supplementary Material, Section S5) with the language task and resting state scans, separately with iGC and dGC features (connections). Prior to analysis of real data, we validated RFE by applying it to estimate key differences in two simulated networks (Supplementary Material Fig. S2 and Fig. S3). RFE accurately identified connections that differed in simulation "ground truth": specifically, differences in fast timescale connections were identified by iGC, and in slow timescale connections by dGC.

When applied to the language task versus resting state fMRI data, RFE identified a small subset of 18(/91) connections based on iGC (Fig. 4A), and an overlapping but non-identical set of 17(/182) connections based on dGC (Fig. 4B); these connections were key to distinguishing task (language)

from resting brain states. Specifically, the highest iGC beta weights, corresponding to the most discriminative iGC connections, occurred among various cognitive control networks, including the anterior and posterior salience networks, the precuneus and the visuospatial network (Fig. 5A). Some of these connections were also detected by dGC. Nevertheless, the highest dGC beta weights occurred for connections to and from the language network, for example from the language network to dorsal default mode network and from the precuneus to the language network (Fig. 5B). Notably, these latter connections were important for classification based on dGC, but not based on iGC. Moreover, iGC identified a connection between the language network and the basal ganglia whereas dGC, in addition, identified the directionality of the connection, as being from the language network to the basal ganglia. In summary, dGC and iGC identified several complementary connections, but dGC alone identified many connections with the language network, indicating that slow processes in this network significantly distinguished language from resting states.

Next, we tested whether estimating dGC after systematically downsampling the fMRI time series would permit identifying maximally discriminative connections at progressively slower timescales. To avoid degradation of GC estimates because of fewer numbers of samples with downsampling (by decimation), we concatenated the different downsampled time series to maintain an identical total number of samples. RFE was applied to GC estimates based on data sampled at different rates: 1.4 Hz, 0.5 Hz, 0.3 Hz and 0.2 Hz corresponding to 1x, 3x, 5x, and 7x of TR (sampling period of 0.72 s, 2.16 s, 3.6 s and 5.04 s), respectively. RFE with dGC identified 17(/182) key connections at each of these timescales (Fig. 4B). Interestingly, some connections manifested in dGC estimates across all sampling rates. For instance, the connection from the precuneus to the language network was important for classification across all sampling rates (Fig. 5C). On the other hand, connections between the language network and various other networks manifested at specific sampling rates only. For instance an outgoing connection from the language network to the basal ganglia manifested only at the 1.4 Hz sampling rate, to the visuospatial network and default mode networks only at 0.5 Hz, to the higher-visual network only at 0.2-0.3 Hz, and an incoming connection from the anterior salience only at 0.2 Hz. None of these connections were identified by the iGC classifier (compare Fig. 5A and 5C). Similar timescale generic and timescale specific connections were observed in other tasks as well (Supplementary Fig. S7). Despite downsampling, RFE accuracies were significantly above chance, although accuracies decreased at lower sampling rates (Fig. 4 lower panels) [20]. Thus, dGC identified distinct connectivity profiles for data sampled at different timescales, without significantly compromising classification performance.

Finally, we sought to provide independent evidence to confirm whether these network connections operated at different timescales. For this, we estimated the average cross coherence (Supplementary Material, Section S7) between the fMRI time series of two connections from the language network that were identified by RFE exclusively at 0.2-0.3 Hz (language to higher visual) and 0.5 Hz (language to visuospatial) sampling rates, respectively (Fig. 5C). Each connection exhibited an extremum in the coherence plot at a frequency which closely matched the respective connection's timescale (Fig. 5D). These findings, from experimental data, provide empirical validation to our simulation results, which indicate that estimating dGC on downsampled data is a tenable approach for identifying functional connections that operate at specific timescales.

## 4 Conclusions

These results contain three novel insights. First, we show that two measures of conditional linear dependence – instantaneous and directed Granger-Geweke causality – provide robust measures of functional connectivity in the brain, resolving over a decade of controversy in the field [23][22]. Second, functional connections identified by iGC and dGC carry complementary information, both in simulated and in real fMRI recordings. In particular, dGC is a powerful approach for identifying reciprocal excitatory-inhibitory connections, which are easily missed by iGC and other same-time correlation based metrics like partial correlations [22]. Third, when processes at multiple timescales exist in the data, our results show that downsampling the time series to different extents provides an effective method for recovering connections at these distinct timescales.

Our simulations highlight the importance of capturing emergent timescales in simulations of neural data. For instance, a widely-cited study [22] employed purely feedforward connectivity matrices with a 50 ms neural timescale in their simulations, and argued that functional connections are not reliably inferred with GC on fMRI data. However, such connectivity matrices preclude the occurrence of

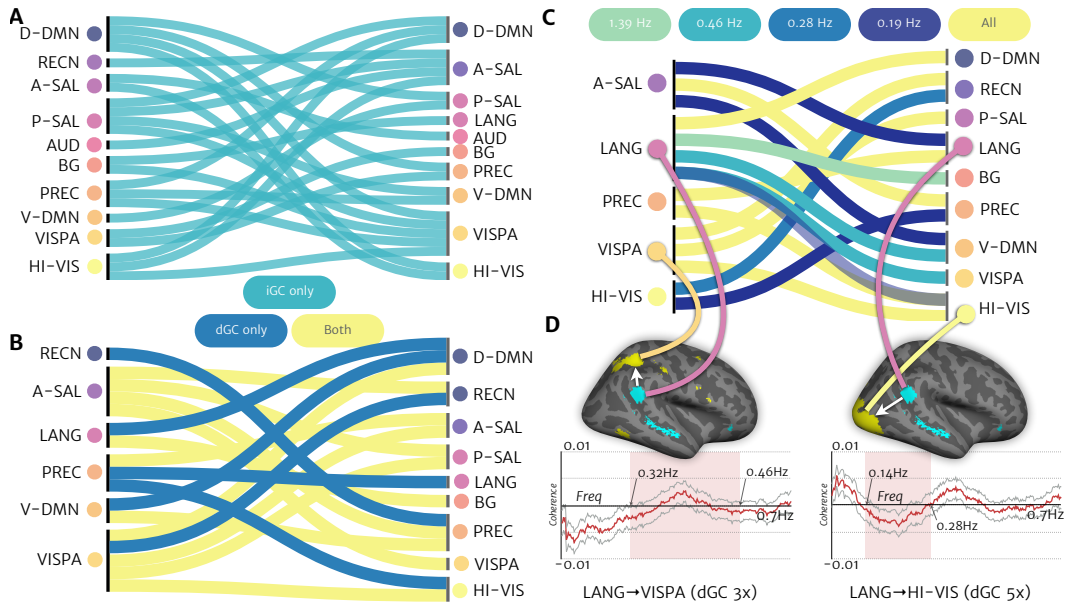

Figure 5: **Connectivity at different timescales.** (A-B) Discriminative connections identified exclusively by iGC (teal), exclusively by dGC (blue), or by both (yellow). Each connection is represented as a band going from a source node on the left to a destination node on the right. (C) (Top) Discriminative connections identified by dGC, exclusively at different sampling intervals (1x, 3x, 5x, 7x TR). (D) (Left) Directed connection between language network and visuospatial network identified by dGC with fMRI data sampled at 0.5 Hz (sampling interval, 3x TR). (Right) Directed connection between language network and higher visual network identified by dGC with fMRI data sampled at 0.3 Hz (sampling interval, 5x TR). (Lower plots) Cross coherence between respective network time series. Shaded area: Frequencies from $F_s/2$ to $F_s$, where $F_s$ is the sampling rate of the fMRI timeseries from which dGC was estimated.

slower, behaviorally relevant timescales of seconds, which readily emerge in the presence of feedback connections, both in simulations [8][15] and in the brain [17][24]. Our simulations explicitly incorporated these slow timescales to show that connections at these timescales could be robustly estimated with GC on simulated fMRI data. Moreover, we show that such slow interactions also occur in human brain networks. Our approach is particularly relevant for studies that seek to investigate dynamic functional connectivity with slow sampling techniques, such as fMRI or calcium imaging.

Our empirical validation of the robustness of GC measures, by applying machine learning to fMRI data from 500 subjects (and 4000 functional scans), is widely relevant for studies that seek to apply GC to estimate directed functional networks from fMRI data. Although, scanner noise or hemodynamic confounds can influence GC estimates in fMRI data [20][4], our results demonstrate that dGC contains enough directed connectivity information for robust prediction, reaching over 95% validation accuracy with averaging even as few as 10 subjects' connectivity matrices (Fig. 3B). These results strongly indicate the existence of slow information flow networks in the brain that can be meaningfully inferred from fMRI data. Future work will test if these functional networks influence behavior at distinct timescales.

**Acknowledgments**. This research was supported by a Wellcome Trust DBT-India Alliance Intermediate Fellowship, a SERB Early Career Research award, a Pratiksha Trust Young Investigator award, a DBT-IISc Partnership program grant, and a Tata Trusts grant (all to DS). We would like to thank Hritik Jain for help with data analysis.

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
