[Supplementary Material · NIPS2017-Final-Supp.pdf]

# Supplementary Material for
# Mapping distinct timescales of functional interactions among brain networks

**Mali Sundaresan**
s.malisundar@gmail.com

**Arshed Nabeel**
arshed@iisc.ac.in

**Devarajan Sridharan**$^*$
sridhar@iisc.ac.in

## S1 Granger Causality estimation using MVAR modelling [5]

Consider the random vector $\mathbf{q}[t]$ whose time evolution is modeled by the following multivariate autoregressive model ("full" model):

$$\mathbf{q}[t] = \sum_{i=1}^{p} A_i \mathbf{q}[t-p] + \mathbf{e}[t] \tag{1}$$

Let $\mathbf{q} = [\mathbf{x}^\top \mathbf{y}^\top \mathbf{z}^\top]^\top$, where $\mathbf{x}$, $\mathbf{y}$ and $\mathbf{z}$ are themselves random vectors. $\mathbf{e}$ is the residual (prediction error) of the full model, with covariance $T$. $T$ can be expressed as comprising the following block matrices:

$$T = \begin{bmatrix} T_{\mathbf{x}} & T_{\mathbf{xy}} & T_{\mathbf{xz}} \\ T_{\mathbf{xy}}^\top & T_{\mathbf{y}} & T_{\mathbf{yz}} \\ T_{\mathbf{xz}}^\top & T_{\mathbf{yz}}^\top & T_{\mathbf{z}} \end{bmatrix} \tag{2}$$

where the submatrices $T_{\mathbf{x}}, T_{\mathbf{y}}$ and $T_{\mathbf{z}}$ are the covariances of the residuals associated with $\mathbf{x}, \mathbf{y}$ and $\mathbf{z}$ (rows), respectively, and $T_{\mathbf{xy}}$ etc are the covariance between the residuals associated with $\mathbf{x}$ and $\mathbf{y}$, etc.

We define the following *reduced* vector autoregressive processes.

$$\mathbf{x}[t] = \sum_{i=1}^{p} A_i^{(\mathbf{xx})} \mathbf{x}[t-p] + \sum_{i=1}^{p} A_i^{(\mathbf{xz})} \mathbf{z}[t-p] + \mathbf{u}[t] \tag{3}$$

$$\mathbf{y}[t] = \sum_{i=1}^{p} A_i^{(\mathbf{yy})} \mathbf{y}[t-p] + \sum_{i=1}^{p} A_i^{(\mathbf{yz})} \mathbf{z}[t-p] + \mathbf{v}[t] \tag{4}$$

$$\tag{5}$$

$\mathbf{u}$ and $\mathbf{v}$ are the residuals of the reduced VAR models, with respective covariance matrices $S_{\mathbf{x}}$ and $S_{\mathbf{y}}$.

The conditional linear dependence between $\mathbf{x}$ and $\mathbf{y}$, conditioned on $\mathbf{z}$, is given by the following equations.

$$\mathcal{F}_{\mathbf{x} \to \mathbf{y} | \mathbf{z}} = \ln \frac{|S_{\mathbf{y}}|}{|T_{\mathbf{y}}|} \tag{6}$$

$$\mathcal{F}_{\mathbf{y} \to \mathbf{x} | \mathbf{z}} = \ln \frac{|S_{\mathbf{x}}|}{|T_{\mathbf{x}}|} \tag{7}$$

$$\mathcal{F}_{\mathbf{x} \circ \mathbf{y} | \mathbf{z}} = \ln \frac{|T_{\mathbf{x}}| \, |T_{\mathbf{y}}|}{|T'|} \tag{8}$$

$$\mathcal{F}_{\mathbf{x}, \mathbf{y} | \mathbf{z}} = \mathcal{F}_{\mathbf{x} \to \mathbf{y}} + \mathcal{F}_{\mathbf{y} \to \mathbf{x}} + \mathcal{F}_{\mathbf{x} \circ \mathbf{y}} \tag{9}$$

$$= \ln \frac{|S_{\mathbf{x}}| \, |S_{\mathbf{y}}|}{|T'|} \tag{10}$$

---

$^*$Corresponding author

where $T' = \begin{bmatrix} T_{\mathbf{x}} & T_{\mathbf{xy}} \\ T_{\mathbf{xy}}^\top & T_{\mathbf{y}} \end{bmatrix}$

## S2  Analytic computation of instantaneous correlations

We consider the case of a two node network and find the analytic solution of correlations between the node time series. The process is simulated as a linear dynamical system of the form:

$$\dot{\mathbf{x}} = A\mathbf{x} + \varepsilon$$

Assuming $\varepsilon$ to be a zero mean Gaussian white noise process with covariance $Q$, it can be shown that the covariance $X$ of time series $\mathbf{x}$ satisfies the Lyapunov equation:

$$AX + XA^T + Q = 0$$

Assuming $A = \begin{bmatrix} -1 & c \\ d & -1 \end{bmatrix}$, it can be shown that the covariance between the components of $\mathbf{x}$ ($X(1,2)$) is a non linear function of $c + d$ and has the same sign as $c + d$ (*lyap* function in Matlab). Importantly, the covariance (and correlation) is 0 for the case of reciprocal excitatory-inhibitory connectivity, for which $c = -d$. This is the reason why iGC and partial correlations, both of which rely on instantaneous correlations failed to capture reciprocal E-I interactions, as shown in Figure 2A (main text).

## S3  fMRI data and time series extraction

We used minimally preprocessed data [6] provided by HCP Consortium[2], for our study. Supplementary Material Fig S8 shows the ids of the 500 subjects who were included in this analysis.

We used a functional 14-network or 90-regional parcellation[3]. Matlab and SPM8[4] were used to extract network time series data from preprocessed scans. The 14 functional networks included here are listed in table Supplementary Material Table S2, along with our abbreviated names for each. We adopted this parcellation, with fewer, more coarse-grained regions, rather than a finer parcellation (e.g. a 274 region functional parcellation [8]) because GC estimates were more reliable when the number of regions was far fewer than the number of timepoints. Both task and resting scans were of sufficient duration ($\sim$200-300 volumes) to permit robust GC estimation. In simulations, we noticed that the magnitude of GC estimates varied based on the number of timepoints used in the estimation. To prevent this difference in number of timepoints from biasing classification performance, we truncated each scan to a common minimum number of time samples for each task and resting scan before estimating GC. Finally, in this parcellation, there were overlapping voxels between some of the networks. To avoid mixing of signals, we assigned each overlapping voxel to the network whose centroid it was closest to.

## S4  Permutation testing

We performed permutation tests for evaluating the statistical significance of classifier performance, using the method outlined in [7]. We conducted these for the language task versus resting state classification, separately for the three metrics (dGC, iGC and fGC).

The first test involved permuting task labels independently for each subject and computing a null distribution of leave-one-out accuracy. We employed 10000 surrogates and confirmed that each of the accuracy values for iGC, dGC and fGC based classifiers (reported in Fig. 3, main text) was significant ($p < 0.0001$).

The second test [7] measures how much of the classification performance is due simply to the differences in the correlation structure of the feature dimensions across resting and task. This was done by permuting the feature dimensions class-wise, and comparing the accuracy of the resultant

classification with the original classification accuracy. We observed that for dGC, RFE accuracy remained similar (80.6%) even after permutation, and over 80% of RFE features were preserved. On the other hand, for iGC, RFE accuracy reduced to 58.8% and only 29% of the features were preserved. These results indicate that iGC relied heavily (and far more than dGC) on dependencies between features for accurate classification.

## S5    Recursive Feature Elimination

Two-level Recursive Feature Elimination (RFE) was implemented as described in previous studies [4][9]. First, the data were divided into $N_1$ (here, 10) folds. Of these, $N_1 - 1$ folds were used as "training" data, and one fold was reserved as "test" data for quantifying the generalization performance of the classifier. Training data were pooled and further divided into $N_2$ (here, 5) folds. The SVM classifier was then trained on $N_2 - 1$ folds (leaving out one fold) and discriminative weights were obtained. The above procedure was repeated $N_2$ times by leaving out each fold, in turn. Average weights were then computed by averaging the absolute values of the discriminative weights across the $N_2$ runs. Next, the feature (connection) contributing the lowest average weight was discarded, and the classifier was trained again with only the retained set of features. This procedure of feature selection and training was repeated until no more features remained. At this stage, the generalization performance for every set of retained features (each RFE level) was assessed using the left out "test" data. The entire procedure was repeated $N_1$ times by leaving out each fold of the original data, in turn, as test data. Final generalization performances and discriminative weights of each RFE level were obtained as the average over $N_1$ folds. We selected the set of connections at the RFE level at which the generalization performance reached an "elbow": the minimum set of connections at which generalization performance dipped dramatically from its maximal level. To identify this elbow we adopted the following procedure: The RFE curves were first smoothed using a moving average filter (length: 5 features). Then the first derivative was computed as a first order difference by subtracting adjacent values. The elbow is defined as the point where the first derivative changes from a positive value to near zero. Therefore we took the left most point at which the first derivative deviated significantly from zero. This corresponded well with our visual estimate for the case of iGC, and dGC at 1x, 3x and 5x down sampling. For the case of the RFE curve with 7x downsampling, the elbow was identified by visual inspection.

## S6    Comparison with partial correlations

We compared the performance of classification based on GC measures with that based on partial correlations (PC). We observed that PC connectiviy performed consistently better than GC connectivity for classifying task from rest (Figure S5B). The better performance of PC could be due to the following reasons. First, estimators based on instantaneous correlations alone are typically less susceptible to noise than those that incorporate lagged correlations. This is due to the fact that the estimation of lagged-covariance is susceptible to errors from noise at multiple time-points. For illustration, consider a VAR(1) generative model $\mathbf{x}(t) = A\mathbf{x}(t-1) + \mathbf{e}(t)$. The lagged covariance matrix is given by $\Sigma_1 = \mathbb{E}\left[\mathbf{x}(t)\mathbf{x}(t-1)^\top\right] = \mathbb{E}\left[(A\mathbf{x}(t-1) + \mathbf{e}(t))\,\mathbf{x}(t-1)\right]^\top = A\mathbb{E}\left[\mathbf{x}(t-1)\mathbf{x}(t-1)^\top\right] + \mathbb{E}\left[\mathbf{e}(t)\mathbf{x}(t-1)^\top\right]$; the variability of the interaction-term $\mathbf{e}(t)\mathbf{x}(t-1)^\top$ contributes to the variance of $\Sigma_1$ in addition to to the variability in computing the instantaneous correlation. Second, information theoretic measures like dGC require sufficient number of samples for reliable estimates and accurate classification, as demonstrated in Figure S6.

## S7    Estimation of coherence between two network time series

We used the Chronux toolbox [3] to compute the coherence between regional time series. We chose two connections that were identified exclusively at two of the downsampling rates viz language to visuospatial network (3x), and language to higher visual network (5x). After mean removal, the time series of language and resting conditions were provided as input to the *coherencyc.m* function. For each subject, we then computed the difference between coherence in the language task minus the coherence in the resting state. The plots in Fig. 5D (lower panel, main text) are obtained by taking the mean and standard error of the coherence (at each frequency) across subjects.

# Supplementary Figures and Tables

Figure S1: **Variation in dGC values at different sampling rates** : A minimal two node network with one positive connection between the nodes was used in this simulation. (A) The analytic solution of estimated dGC [2] for the connection at three different process timescales - 50 ms (blue), 500 ms(red) and 2 s (black). Note that the peak of each curve matches the process timescale (dotted line). (B) Same as in A, but when the time series were also filtered with a hemodynamic response function (average of 50 simulations). As before, peak dGC values occur close to the sampling interval.

Figure S2: **Validation of Recursive Feature Elimination (RFE) using simulations**. (A) Two networks used in the simulations. The first three nodes of the networks have slow (2s) decays and last three have fast (50 ms) decays. 100 fMRI time series were simulated and sampled at a TR=2 s. (B) (Bottom) RFE analysis was done on the iGC and dGC measures estimated from the time series and the optimal number of features was identified based on the elbow point of the generalization performance curve. (Top) The optimal identified features for both iGC (left) and dGC (right) correspond closely to the connections in the "ground truth" matrix. iGC could not estimate reciprocal connections whereas dGC could not estimate connections at a timescale much faster than the sampling rate.

Figure S3: **Validation of RFE using simulations at different sampling rates**. (A) Two networks used in the simulations. The first two nodes of the both networks have fast (200 ms) decay while the next two nodes have slow (3s) decay. There is one fast and slow connection in each, but with different magnitudes. As before, 100 simulated fMRI time series were generated. Time series were sampled at two different sampling time periods – TR=200 ms and TR=3 s – and functional connectivity estimated with dGC. (B) In each case RFE estimated precisely one connection at a timescale corresponding to the respective TR (200 ms, left or 3 s, right). Other conventions are the same as in Figure S2B.

Figure S4: **Classification accuracies based on GC for task vs. rest (all tasks)** Leave-one-out classification accuracies for distinguishing seven different task-datasets from resting-state (pairwise classification). (Top) Classification accuracy with dGC, iGC and fGC networks as features, based on the 14-network parcellation. (Bottom) Same as top panel, but classification accuracies for the 90-region parcellation. Other conventions are the same as in Figure 3A, main text

Figure S5: **Connectivity estimation with partial-correlations.** (A) The partial-correlation matrix and the reconstructed network, computed from simulated timeseries generated by Network H (Fig. 1A, main text). Conventions are the same as in Fig. 1A of main text. (B) Accuracy of classification based on PC-features, for discriminating resting state from each of the seven tasks. (C) RFE curves, with classification accuracy as a function of remaining features, for PC-based classification. Other conventions are the same as in Figure 4, main text

Figure S6: **Variation of classification accuracy for dGC connectivity with number of timepoints.** Classification accuracy, based on dGC, as a function of the length of the timeseries used for estimating the dGC matrix. (A) Classification accuracy for networks estimated from simulated data. Timeseries were simulated with two different "ground truth" networks (Fig. 1A-B) and used for dGC estimation. (B) Classification accuracies with increasing number of timepoints in real fMRI data, for discriminating resting state and the language task.

Figure S7: **Connectivity at different timescales – other tasks.** Discriminative connections identified by dGC for the (A) working memory and the (B) social tasks. Connections exclusive to different sampling-rates, and ones common to all sampling-rates are shown. Other conventions are the same as in Figure 5, main text.

| Task | Description |
|------|-------------|
| WORKING MEMORY | A version of the n-back task, requires holding sequences in memory. |
| LANG | Participants listen to brief stories and in the end are asked questions about the stories. |
| MOTOR | Involves moving fingers, toes or tongue in response to visual cues. |
| SOCIAL | Participants are shown video clips of moving objects, and are asked to make judgments about social interactions among them. |
| GAMBLING | A guessing game where participants guess the number on a hidden card. |
| RELATIONAL | Participants are asked to identify similarities among objects of varying shapes and textures. |
| EMOTIONAL | Participants make comparisons between images of faces, and make judgments about the emotion portrayed. |

Table S1: Descriptions of the seven tasks [1].

| Short Name | Functional Network |
|------------|-------------------|
| A-SAL | Anterior Insula / Dorsal ACC (Anterior Salience Network) |
| AUD | Auditory Network |
| BG | Basal Ganglia Network |
| D-DMN | PCC / MPFC (Dorsal Default Mode Network) |
| LANG | Language Network |
| LECN | Left DLPFC / Parietal (Left Executive Control Network) |
| SENMOT | Sensorimotor Network |
| P-SAL | Posterior Insula (Posterior Salience Network) |
| PREC | Precuneus Network |
| PR-VIS | Primary Visual Network |
| HI-VIS | Higher Visual Network |
| RECN | Right DLPFC / Parietal (Right Executive Control Network) |
| V-DMN | Retrosplenial Cortex / Medial Temporal Lobe (Ventral Default Mode Network) |
| VISPA | Intraparietal Sulcus / Frontal Eye Fields (Visuospatial Network) |

Table S2: 14 functional networks, along with their short names [10].

| | | | | | | | | | |
|---|---|---|---|---|---|---|---|---|---|
| 100206 | 130013 | 103414 | 141422 | 148941 | 154431 | 105923 | 164636 | 172130 | 177241 |
| 102008 | 130316 | 135528 | 141826 | 149236 | 154532 | 159239 | 164939 | 172332 | 177645 |
| 123824 | 130417 | 135730 | 142828 | 149337 | 154734 | 159340 | 165032 | 172433 | 178142 |
| 123925 | 130619 | 135932 | 143325 | 149539 | 105216 | 159441 | 165638 | 172534 | 178243 |
| 124220 | 130821 | 136227 | 104012 | 149741 | 154835 | 159744 | 165840 | 172938 | 178647 |
| 124422 | 102816 | 136732 | 144125 | 149842 | 154936 | 159946 | 106521 | 107321 | 178748 |
| 124624 | 130922 | 136833 | 144731 | 150524 | 155231 | 160123 | 166438 | 173334 | 178849 |
| 124826 | 131217 | 137027 | 144832 | 105014 | 155635 | 160729 | 166640 | 173435 | 178950 |
| 125525 | 131419 | 137128 | 145127 | 150625 | 155938 | 160830 | 167036 | 173536 | 108121 |
| 126325 | 131722 | 137229 | 145834 | 150726 | 156031 | 106016 | 167238 | 173637 | 179245 |
| 126628 | 131823 | 137633 | 146129 | 150928 | 156233 | 161327 | 167743 | 173738 | 179346 |
| 102311 | 131924 | 103515 | 146331 | 151223 | 156334 | 161630 | 168240 | 173839 | 180129 |
| 127327 | 132017 | 137936 | 104416 | 151425 | 156435 | 161731 | 168341 | 173940 | 180432 |
| 127630 | 133019 | 138231 | 146432 | 151526 | 156536 | 162026 | 168745 | 174437 | 180735 |
| 127933 | 103111 | 138534 | 146533 | 151627 | 105620 | 162228 | 107018 | 175035 | 180836 |
| 128026 | 133625 | 138837 | 146937 | 151728 | 156637 | 162329 | 169444 | 107422 | 180937 |
| 128127 | 133827 | 139233 | 147030 | 151829 | 157336 | 162733 | 169747 | 175237 | 181131 |
| 128632 | 133928 | 139637 | 147737 | 152831 | 157437 | 162935 | 169949 | 175338 | 108222 |
| 128935 | 134021 | 139839 | 148032 | 105115 | 157942 | 163129 | 170631 | 175439 | 181232 |
| 129028 | 134223 | 140117 | 148133 | 153025 | 158035 | 163331 | 170934 | 175742 | 181636 |
| 129129 | 134324 | 140319 | 148335 | 153227 | 158136 | 106319 | 171330 | 176037 | 182436 |
| 102513 | 134425 | 103818 | 148436 | 153429 | 158338 | 163432 | 100408 | 176239 | 182739 |
| 129331 | 134728 | 140824 | 100307 | 153631 | 158540 | 163836 | 171532 | 176441 | 183034 |
| 129634 | 134829 | 140925 | 104820 | 153833 | 158843 | 164030 | 171633 | 176542 | 183337 |
| 129937 | 135225 | 141119 | 148840 | 154229 | 159138 | 164131 | 172029 | 107725 | 185139 |
| 185341 | 192136 | 198653 | 204622 | 211316 | 231928 | 283543 | 310621 | 358144 | 395756 |
| 108323 | 192439 | 109830 | 205119 | 211417 | 233326 | 112314 | 311320 | 113619 | 395958 |
| 185442 | 192540 | 198855 | 205220 | 211720 | 236130 | 284646 | 316633 | 361234 | 397154 |
| 185846 | 192641 | 199453 | 205725 | 211922 | 237334 | 285345 | 316835 | 361941 | 397760 |
| 185947 | 192843 | 199655 | 110613 | 212015 | 111716 | 285446 | 317332 | 365343 | 397861 |
| 186141 | 193239 | 199958 | 205826 | 212116 | 239944 | 286650 | 318637 | 366042 | 406432 |
| 186444 | 109123 | 200008 | 206222 | 212217 | 245333 | 287248 | 112920 | 366446 | 406836 |
| 187143 | 194140 | 200109 | 207123 | 212318 | 246133 | 289555 | 321323 | 371843 | 412528 |
| 187345 | 194645 | 200210 | 207426 | 212419 | 248339 | 293748 | 322224 | 377451 | 114217 |
| 187547 | 194746 | 200311 | 208024 | 212823 | 249947 | 295146 | 329440 | 378857 | 414229 |
| 187850 | 194847 | 110007 | 208125 | 111413 | 250427 | 297655 | 330324 | 379657 | 415837 |
| 188347 | 195041 | 200614 | 208226 | 213421 | 250932 | 112516 | 333330 | 380036 | 422632 |
| 108525 | 195445 | 201111 | 208327 | 214019 | 251833 | 298051 | 334635 | 381038 | 424939 |
| 188448 | 195849 | 201414 | 111009 | 214221 | 255639 | 298455 | 336841 | 381543 | 429040 |
| 188549 | 195950 | 201515 | 209127 | 214423 | 256540 | 299154 | 339847 | 382242 | 432332 |
| 188751 | 196144 | 201818 | 209228 | 214524 | 101006 | 300618 | 341834 | 385450 | 433839 |
| 189349 | 100610 | 202113 | 209329 | 214726 | 112112 | 303119 | 346137 | 386250 | 436239 |
| 189450 | 109325 | 202719 | 209834 | 217126 | 257542 | 303624 | 113215 | 387959 | 436845 |
| 190031 | 196346 | 203418 | 209935 | 217429 | 257845 | 304020 | 346945 | 389357 | 441939 |
| 191033 | 196750 | 110411 | 210011 | 220721 | 263436 | 304727 | 348545 | 390645 | 101107 |
| 191336 | 197348 | 203923 | 210415 | 111514 | 268749 | 305830 | 352132 | 391748 | 114318 |
| 108828 | 197550 | 204016 | 210617 | 221319 | 268850 | 307127 | 352738 | 393247 | 114419 |
| 191841 | 198249 | 204319 | 211114 | 224022 | 270332 | 308129 | 353740 | 113922 | 114621 |
| 191942 | 198350 | 204420 | 211215 | 227432 | 275645 | 308331 | 355239 | 393550 | 114823 |
| 192035 | 198451 | 204521 | 111312 | 228434 | 280739 | 309636 | 356948 | 395251 | 114924 |

Figure S8: **HCP subject identifiers.** Unique identifiers for the 500 subjects from the HCP database [6] that were analyzed in this study.

## Footnotes

[2]http://www.humanconnectome.org/

[3]http://findlab.stanford.edu/functional_ROIs.html

[4]www.fil.ion.ucl.ac.uk/spm/software/spm8/