[Reviews · NeurIPS 2017]

Reviewer 1



# Main paper Overall an interesting read but the novelty is not so clear to me. ## Section 2.1 It is not specified how the MVAR model order (p in S1 eqs 1-3) is selected, while this is an important hyperparameter. In particular for cross-validation experiments it needs to be trained in-fold. Because the paper explicitly resamples fMRI time series and aims for interpretability, it would be useful to mention how the max lag changes with resampling so that it can be made explicit that the time scale of the lag is in range with plausible causality in brain circuits and what we already know about long-memory processes. Figure 1 is very nice but for panels A and B the stripe plot (bottom right of each panel) is quite redundant given that there are already two other representation of the networks in the same panel. ## Section 3.1 Around line 171: please explicitly mention your feature space dimensions here, easier to understand if we don't have to look for this later. Here and in section 3.2 the feature spaces generated have different dimensions (91, 182, 273) and more importantly probably different correlation structures and sparsity patterns. There is no mention here of the type of regularization that is used, nor of how that hyperparameter is tuned. ## Section 3.2 For rest vs task, the accuracy reported (single-subject case) actually seems quite low (maybe the atlas choice?) - but it's hard to say without comparison. If you want to make an argument that iGC and dGC are good representations of the data because they are sensitive to task differences, then it would be good to compare with cruder techniques like classical correlation and partial correlation (e.g. with Whittaker-style inversion). lines 198-199 : how exactly are matrices 'averaged across several folds of S' ? Unclear to me how that sits in the cross-validation scheme. lines 206 - 'across as few as 10 subjects' - that in fact seems to me like a lot - you are losing individual predictions, so if that's what it takes to stabilise the estimator it's in fact not very suitable for single-subject prediction I would argue. Again, hard to say without comparison to other connectivity estimators. # Supplementary ## Section S1 Covariance matrices in equation 4 should be described explicitly - e.g. $\Sigma_1$ is the reduced covariance matrix, and $\Sigma_2$ is the full covariance matrix - by line 7 the matrices with subscript 2 have not been defined. ## Section S2 line 13 - 'off-diagonal correlation value' should be 'off-diagonal covariance value'.

Reviewer 2



The paper deals with a controversial problem related with the use of Granger-Geweke causality (GC) to fMRI data. This paper proposes a classification task as an indirect validation of the discriminant power of functional connectivity features based on GC. In particular, they define a set of features used in a machine learning approach (SVM classifier + Recursive feature elimination (RFE) for feature selection) and analyze the classification performance and the features. These can result in a characterization of the brain functional networks. The work is well motivated, including a broad introduction about the issue in the first section and a proper presentation of the different concepts related with fMRI (which I am not confident to completely understand). The machine learning approach has sense for me and serve to show that the features are significant to distinguish task versus rest conditions. I have only one concern about the RFE method application which is the fact that the number of samples (500 subjects) is not much larger than dimensions of the feature vector (~182). It could be interesting to validate the obtained result by using permutation testing. As a minor comment, the last section of the paper should be conclusions.

Reviewer 3



The authors apply instantaneous and lag-based Granger-Geweke causality (iGC and dGC) to infer brain network connections at distinct timescales from functional MRI data. With simulated fMRI data and through analysis of real fMRI recordings, they presented the following novel insights. First, iGC and dGC provide robust measures of brain functional connectivities, resolving over long-term controversy in the field. Second, iGC and dGC can identify complementary functional connections, in particular, the latter can find excretory-inhibitory connections. Third, downsampling the time series to different extents provides an efficient method for recovering connections at different timescales. Clarity: The paper is well organised and clearly written with nice figures. The result of real data analysis may be difficult to understand without background knowledge in neuroscience. Significance: I think that the insights written in this paper have significant impact on researchers working on this topic. Recently analyses of resting-state fMRI data have led to many interesting results in neuroscience. However, most publications employed correlation matrices for measuring functional connectivities. So, if iGC and dGC would become alternative tools, we could extract/grasp multi timescale processing in the brain. In order to convince more researchers of practical usefulness of iGC and dGC, it would be necessary to compare with existing techniques further and to work on more real fMRI examples. On the other hand, the scope of this paper is rather narrow and impacts on other research topics in the NIPS community are rather limited. Correctness: Since the authors applied existing connectivity measures, I don't expect any technical errors. Originality: I feel that a certain amount of researchers are sceptical about applications of GC measures to fMRI data. I have not seen the messages of this paper which may resolve over long-term controversy in the field. To sum up, the novel insights derived from analysis of simulated and real fMRI data with GC measures are interesting and could have potential to advance the research field. Since further real-world examples would be preferable to convince other researchers in the field, I would say that this paper is weak accept. There are a few minor comments. - p.3 Eq.(2): I know a MVAR model with variable time lag for each connection which is determined by DTI fibre length. The simulation model in Section 2 assumes variable timescale at each node. I wonder whether we can set timescale of each connections separately at a predetermined value in this formulation or we have to further extend the model in Section 2 to construct more realistic model incorporating knowledge from structural MRI data. - Section 3: I guess that the language task could contain slow brain processing and might be a good example to show multi timescale network structures by fMRI downsampling. I wonder whether there exist other tasks in HCP which we can see similar results.